

# Performance of tree-building methods using a morphological dataset and a well-supported Hexapoda phylogeny

Felipe Francisco Barbosa[1], José Ricardo M. Mermudes[1] and Claudia A. M. Russo[2]

[1] Zoology, Universidade Federal do Rio de Janeiro, Rio de Janeiro, Rio de Janeiro, Brazil
[2] Genetics, Universidade Federal do Rio de Janeiro, Rio de Janeiro, Rio de Janeiro, Brazil

Corresponding author
Claudia A. M. Russo,
claurusso@hotmail.com

## ABSTRACT

Recently, many studies have addressed the performance of phylogenetic tree-building methods (maximum parsimony, maximum likelihood, and Bayesian inference), focusing primarily on simulated data. However, for discrete morphological data, there is no consensus yet on which methods recover the phylogeny with better performance. To address this lack of consensus, we investigate the performance of different methods using an empirical dataset for hexapods as a model. As an empirical test of performance, we applied normalized indices to effectively measure accuracy (normalized Robinson–Foulds metric, nRF) and precision, which are measured *via* resolution, one minus Colless' consensus fork index (1-CFI). Additionally, to further explore phylogenetic accuracy and support measures, we calculated other statistics, such as the true positive rate (statistical power) and the false positive rate (type I error), and constructed receiver operating characteristic plots to visualize the relationship between these statistics. We applied the normalized indices to the reconstructed trees from the reanalyses of an empirical discrete morphological dataset from extant Hexapoda using a well-supported phylogenomic tree as a reference. Maximum likelihood and Bayesian inference applying the k-state Markov (Mk) model (without or with a discrete gamma distribution) performed better, showing higher precision (resolution). Additionally, our results suggest that most available tree topology tests are reliable estimators of the performance measures applied in this study. Thus, we suggest that likelihood-based methods and tree topology tests should be used more often in phylogenetic tree studies based on discrete morphological characters. Our study provides a fair indication that morphological datasets have robust phylogenetic signal.

## INTRODUCTION

As organisms inherit their genome from their ancestors, phylogenetic trees are key to predictability in the Life Sciences, and they are routinely used in most biological areas (*Felsenstein, 2004a*). Using molecular datasets, likelihood-based methods (maximum

likelihood and Bayesian inference) are widely used in phylogenetic analyses (*Chen, Kuo & Lewis, 2014*; *Felsenstein, 2004a*), with a few genetic markers (*Sanger, Nicklen & Coulson, 1977*) or genomic data (*Young & Gillung, 2020*). Nevertheless, the performance of tree-building methods using morphological data is somewhat controversial (but see *Felsenstein, 1978*; *Lewis, 2001*) despite the predominance of the parsimony method (*Farris, 1983*; *Kitching et al., 1998*). As morphological data are the only type available in palaeontology, it is important to evaluate methods using morphology (*e.g.*, *Goloboff, Torres & Arias, 2018*; *Puttick et al., 2018*; *Schrago, Aguiar & Mello, 2018*; *Smith, 2019*).

Computer simulations may be used to estimate the performance of tree-building methods, as the user selects the actual phylogenetic tree by which simulated lineages evolve. Hence, in those cases, there is a known true tree that is comparable to those reconstructed by distinct methods which can be then evaluated. There have been many recent morphology-based simulations that indicate that the Bayesian inference outperforms both maximum likelihood and maximum parsimony in recovering the true tree (*O'Reilly et al., 2016*, *2017*, *2018*; *Puttick et al., 2017a*, *2017b*, *2018*; *Wright & Hillis, 2014*). Conversely, also using simulations, *Smith (2019)* concluded that implied-weights maximum parsimony and Bayesian inference seem to converge as the amount of morphological data increases. Simulations, however, often rely on an unrealistic combination of parameter values, particularly when using morphological data (*Goloboff et al., 2019*; *O'Reilly et al., 2018*). Some researchers suggest, for instance, that the traditional k-state Markov (Mk) model (*Lewis, 2001*) may be biased (*e.g.*, *Goloboff, Torres & Arias, 2017*, *2018*) towards likelihood-based methods. Naturally, model-based simulated datasets tend to oversimplify evolutionary processes, potentially leading to biased results (*Goloboff, Torres & Arias, 2017*; *O'Reilly et al., 2016*; *Wright & Hillis, 2014*).

To evaluate the performance of tree-building methods using morphological data, one possible alternative is to use a reference tree, that is, a tree that is well-supported by fossils, molecules, phylogenomics and morphology (*e.g.*, *Miyamoto & Fitch, 1995*). This popular approach is partially derived from classical congruence studies and relies on the premise that the congruent or reference topology is the best estimate for the true topology of a group; thus, by comparing reference and inferred trees, we would be able to measure phylogenetic accuracy (*Hillis, 1995*).

Therefore, in this study, to evaluate the performance of tree-building methods, we used a discrete morphological dataset (*Beutel & Gorb, 2001*) to test maximum parsimony, maximum likelihood, and Bayesian inference in recovering a phylogenomic extant Hexapoda topology (*Misof et al., 2014*). We compared reconstructed and reference trees using a topological distance metric that measures accuracy (*Day, 1986*; *Penny, Foulds & Hendy, 1982*; *Robinson & Foulds, 1981*). Additionally, we analysed other metrics, such as precision (measured *via* resolution) and the relationship with support measures (see section "Metrics and indices" in Methods for more on our use of "accuracy").

## MATERIALS AND METHODS

### Data matrix, reference tree, and phylogenetic reconstructions

We aimed to study the performance of the metrics of accuracy and precision (measured *via* resolution) and their relationship with support measures (*Brown et al., 2017*; *Smith, 2019*). As previously reported, we used the reference topology resulting from the phylogenomic analysis of 1,478 protein-coding genes (1K Insect Transcriptome Evolution Project, 1KITE, https://1kite.org; *Misof et al., 2014*; Fig. 1). We selected this topology because many recent studies have used this tree as a reference for Hexapoda (*e.g.*, *Beutel et al., 2017*; *Boudinot, 2018*; *Moreno-Carmona, Cameron & Quiroga, 2021*; *Kjer et al., 2016a, 2016b*). Additionally, other reconstructed topologies (*e.g.*, *Peters et al., 2014*; *Song et al., 2016*; *Thomas et al., 2020*; *Wipfler et al., 2019*) and fossil taxa (*Wolfe et al., 2016*) were congruent with this topology, rendering a high degree of phylogenetic confidence (*Hillis, 1995*; *Miyamoto & Fitch, 1995*).

The original taxonomic names (in the data matrix and reference tree) were revised according to the most recent information (*Beutel et al., 2017*; *Grimaldi & Engel, 2005*; *Kjer et al., 2016a, 2016b*; see File S1). A summary flowchart showing the operational steps of the present study is shown in Fig. 2. Our phylogenies were reconstructed using an empirical discrete morphological dataset from extant Hexapoda, (*Beutel & Gorb, 2001*; 115 characters, of which 98 are parsimony informative and seven are constant; equal-weights maximum parsimony ensemble consistency index = 0.697; equal-weights maximum parsimony ensemble retention index = 0.765 (*Farris, 1989*); G1 statistic = −0.716 (*Hillis & Huelsenbeck, 1992*; *Sokal & Rohlf, 1981*)), to test the performance of the tree-building methods in recovering the selected well-supported phylogenomic topology (*Misof et al., 2014*).

The following phylogenetic tree-building methods were used in this study: (1) Bayesian inference (BI; *Rannala & Yang, 1996*; *Yang & Rannala, 1997*), (2) maximum likelihood (ML; *Felsenstein, 1973*, *1981*), (3) unordered (nonadditive) equal-weights maximum parsimony (EW-MP; *Farris, 1983*; *Fitch, 1971*), and (4) unordered (nonadditive) implied-weights maximum parsimony (IW-MP; *Goloboff, 1993*). In the latter, several values for the constant (K parameter) were used (2, 3, 5, 10, and 20) for the homoplasy concavity function, which modifies the weights of characters, downweighting more homoplastic characters. Hence, seeking to avoid a possible bias, we tested several values following the procedure of *Smith (2019)*.

The k-state Markov (Mk) model (*Lewis, 2001*) is, to a certain degree, a generalization of the *Jukes & Cantor (1969)* model (JC69) for discrete morphological data applied to k (unordered)-state characters, assuming evolution *via* a stochastic Markovian process. Using the Mk model, we can assign a $2 \times 2$ rate matrix (k = 2) for binary characters or a higher dimensionality rate matrix (k > 2) for multistate characters. Hence, the dimensionality of the Mk matrices exhibits variability among characters. In the case of the JC69 model, the process is always modelled in a $4 \times 4$ rate matrix for all characters. Therefore, the JC69 model could be seen as a special case of the Mk model for k = 4 (see *Felsenstein, 1973*; *Lewis, 2001*; *Pagel, 1994* for more details).

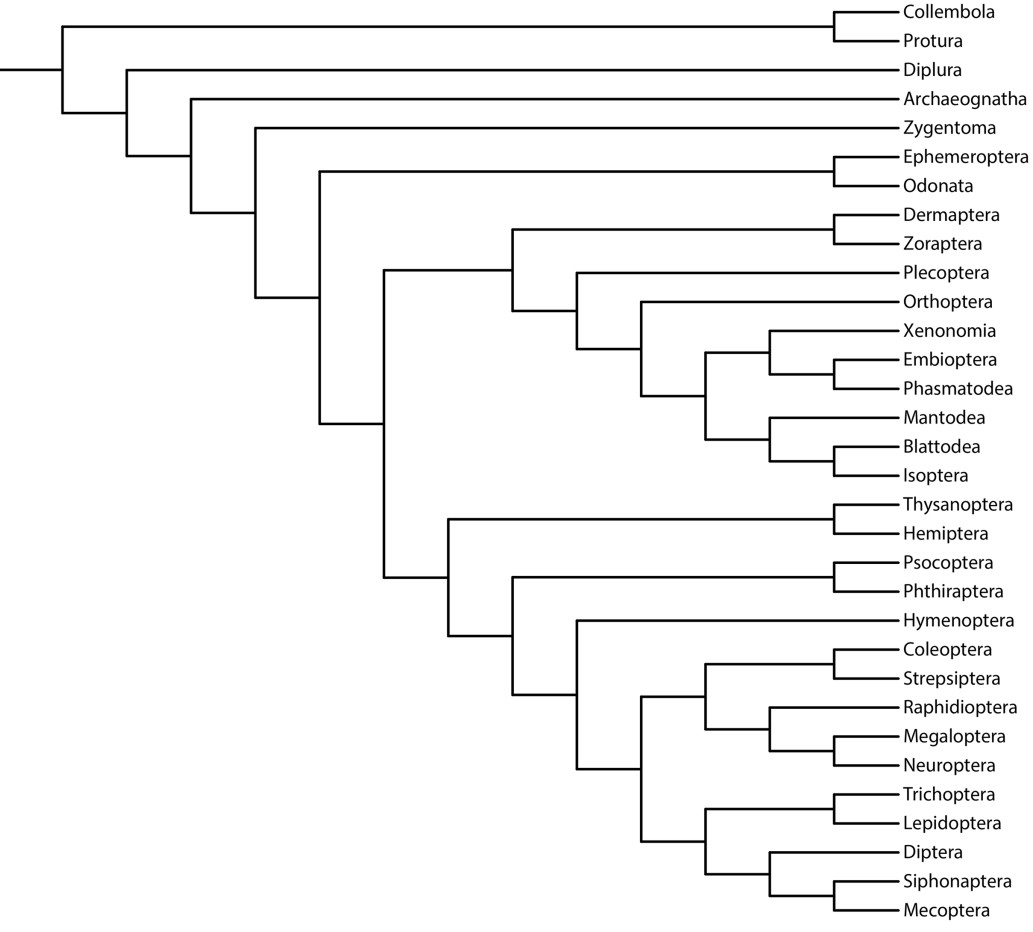

**Figure 1 Phylogeny of Hexapoda.** Phylogeny of the subphylum Hexapoda used in the present work as a well-supported reference cladogram. This phylogenetic tree was pruned from the phylogenomic analysis of 1,478 protein-coding genes, 1KITE project, https://1kite.org (*Misof et al., 2014*; see also *Beutel et al., 2017* and *Kjer et al., 2016a*, *2016b*). Operational adaptations made in the terminal groups are detailed in File S1.               

In the present study, the Mk model was modelled with and without a discrete gamma distribution (Mk+G and Mk, respectively) to account for the heterogeneity rates across sites/characters (*Jin & Nei, 1990*; *Uzzell & Corbin, 1971*; *Yang, 1993*, *1994*). PAUP* 4.0a169 software (*Swofford & Bell, 2017*) was used for EW-MP and IW-MP; IQ-Tree 2 software (*Minh et al., 2020*; *Nguyen et al., 2015*) was used for ML; and MrBayes 3.2.7a software (*Ronquist et al., 2012*) was used for BI.

In the case of the BI runs, chain convergence of the posterior distribution was checked *via* Tracer 1.7.1 software (*Rambaut et al., 2018*), evaluating the parameters: for standard deviations of split frequencies, the minimum threshold of 0.01 was adopted; for minimum and average values of the effective sample size (ESS; *Ripley, 1987*), the recommended minimum threshold of 200 was adopted; and the potential scale reduction factor (PSRF; *Gelman & Rubin, 1992*) approached 1.000.

Statistical support for branches (*Wróbel, 2008*) was generated by (1) nonparametric bootstrap (*i.e.*, standard bootstrap) for EW-MP and IW-MP; this resampling method was

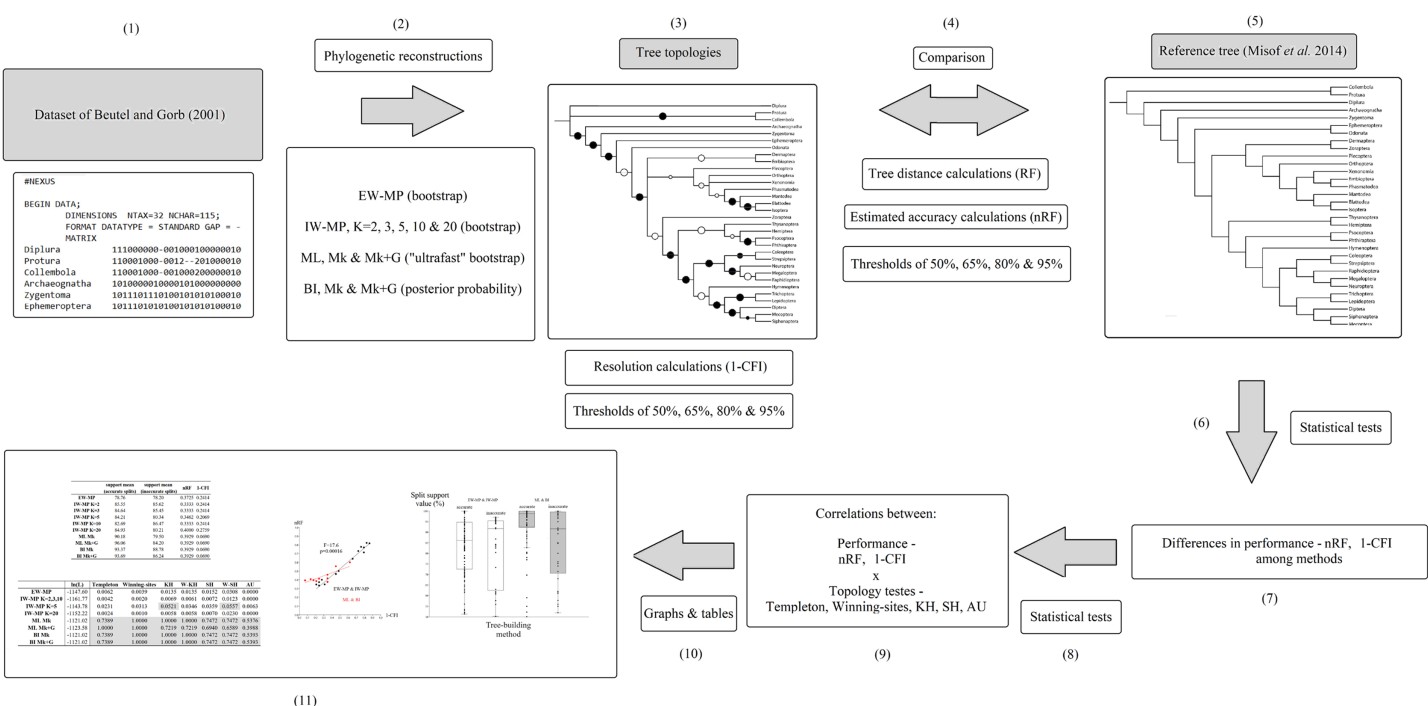

**Figure 2  Flowchart.** Summary flowchart showing the operational steps of the present study. Steps (1) to (3) and (5) are characterized and detailed in the section "Data matrix, reference tree, and phylogenetic reconstructions". Steps (3) to (5) are characterized and detailed in the section "Metrics and indices". Steps (6) to (11) are characterized and detailed in the section "Statistical and tree topology tests".

adapted by *Felsenstein (1985)* from the original proposal of *Efron (1979)*; (2) "ultrafast" bootstrap (UFBoot; *Hoang et al., 2018*; *Minh, Nguyen & von Haeseler, 2013*) for maximum likelihood analyses; and (3) posterior probability (PP) of Bayesian inference (*Yang & Rannala, 1997*). Note that the application of the "ultrafast" bootstrap (developed for likelihood analyses) has increased recently, since it is not just an ordinary faster improvement compared to the standard bootstrap but is also reportedly a more accurate resampling method for maximum likelihood (*Hoang et al., 2018*). Therefore, the support chosen and applied here is consistent with traditional and current phylogenetic practices among phylogenetic tree-building methods (*Hoang et al., 2018*; *Minh, Nguyen & von Haeseler, 2013*; *Minh et al., 2020*; *Nguyen et al., 2015*).

All the resulting trees from all the phylogenetic methods were summarized by the majority-rule consensus tree method (*Margush & McMorris, 1981*) for nonparametric bootstrap, "ultrafast" bootstrap, and posterior probability. The majority-rule consensus tree was used to summarize the BI results, considering that this strategy is more accurate (also when the trade-off with precision is considered) than the alternatives, maximum clade credibility consensus tree (MCC) and maximum *a posteriori* tree (MAP) (*Holder, Sukumaran & Lewis, 2008*; *O'Reilly & Donoghue, 2018*; *Rannala & Yang, 1996*).

Next, among all methods, the groups were collapsed using four different support thresholds (1-α) (majority-rule consensus = 50%, 65%, 80%, and 95%). *Brown et al. (2017)* demonstrated that accuracy and precision measures should preferentially be calculated for
trees with comparable support thresholds (and not in optimal point estimate trees, such as an optimal maximum likelihood tree). If different support threshold trees are compared directly (*e.g.*, an majority-rule consensus compared to a 95% support threshold tree), precision is overestimated (in the majority-rule consensus) or underestimated (in the 95% support threshold tree), and thus, the results are misleading (see also *Alfaro, Zoller & Lutzoni, 2003*; *Berry & Gascuel, 1996*). Finally, the resulting cladograms were visualized using the Interactive Tree of Life online platform (iTOL) v5 (*Letunic & Bork, 2021*).

## Metrics and indices

To assess the performance of the phylogenetic tree-building methods tested in this study, the three metrics used were accuracy, precision, and statistical support measures. Accuracy measures the degree of the "true" evolutionary relationships recovered (*Hillis & Bull, 1993*; *Hillis, 1995*). We acknowledge a degree of uncertainty in our reference tree since we are using a well-supported empirical tree; hence, the terms accuracy and true/false are not applicable in the strict sense (*i.e.*, our accuracy measure is a reasonable proxy). However, we decided to use it due to the lack of a more appropriate term (references that use the term in this sense or in a similar sense: *Cunningham, 1997*; *Hipp, Hall & Sytsma, 2004*; *Russo, Takezaki & Nei, 1996*; *Seixas, Paiva & Russo, 2016*).

Precision is a more straightforward measurement, which in this case corresponds to the resolution of a phylogeny (*i.e.*, the degree of evolutionary relationships recovered) (*Brown et al., 2017*). Finally, the statistical support measures correspond to the degree of confidence, or uncertainty, of interior branches in a phylogenetic tree (*Wróbel, 2008*), which in turn directly impacts the precision; this happens if we apply a support threshold, which in turn generates a trade-off with accuracy (*Holder, Sukumaran & Lewis, 2008*; *Brown et al., 2017*; *O'Reilly & Donoghue, 2018*; *Smith, 2019*). On this subject, collapsing poorly supported clades into soft polytomies often improves the overall accuracy, and these poorly supported clades should not be considered reliable in general (see *O'Reilly & Donoghue, 2018*). Here, we standardized the latter metric simply as nodal "support", which includes phylogenetic resampling methods, such as the nonparametric bootstrap and the "ultrafast" bootstrap, as well as the posterior probability of Bayesian inference (*Alfaro, Zoller & Lutzoni, 2003*; *Hillis & Bull, 1993*; *Soltis & Soltis, 2003*).

We evaluated the performance (accuracy and precision and their relationship with support measures) of different categories of tree-building methods, such as parsimony *vs* likelihood-based methods, since these measures are complementary (*Mackay, 1950*; *Smith, 2019*). The topological distance between two trees has traditionally been calculated by the Robinson–Foulds metric (RF), also known as the "Robinson–Foulds distance", "symmetric difference", or "partition distance" (*Bourque, 1978*; *Penny, Foulds & Hendy, 1982*; *Robinson, 1971*; *Robinson & Foulds, 1981*), with a widely applied modification (*Rzhetsky & Nei, 1992*) that enables calculation in multifurcating trees.

The Robinson–Foulds metric may be used to estimate accuracy when comparing a reconstructed tree with a reference tree (*Hillis, 1995*). As the distance (RF) range is not 0 to 1, a comparison between different metrics (*e.g.*, a comparison with a precision metric) is inappropriate without adequate normalization. The most suitable normalization, termed

"symmetric difference", was proposed by *Day (1986)*, in which the Robinson–Foulds absolute value (RF) is divided by the sum of the total number of resolved (not polytomous) nontrivial splits in each tree. In this study, it is simply termed the "normalized Robinson–Foulds metric" (nRF). The RF absolute values were calculated using the "treedist" of the PHYLIP 3.698 program package, PHYLogeny Inference Package (*Felsenstein, 2013*), and subsequently, these values were normalized *via* the procedure mentioned above.

Regarding the precision (measured *via* resolution), we calculated the ratio of the number of unresolved nontrivial splits or polytomic splits (NRS) to the number of possible nontrivial splits (PS), which also corresponds to one minus Colless' consensus fork index (CFI) (*Colless, 1980*, *1981*). We used this complementary measure in relation to the CFI index, which ranges from 0 to 1, because a perfectly resolved phylogeny would have a value of zero, whereas a totally polytomic tree would have a value of one. Thus, the values of phylogenetic resolution (1-CFI) can be compared on the same scale as the values of phylogenetic accuracy (nRF).

Additionally, to further explore phylogenetic accuracy and support measures (*e.g.*, *Anisimova & Gascuel, 2006*; *Anisimova et al., 2011*; *Berry & Gascuel, 1996*), we calculated other statistics: (1) the true positive rate (or statistical power), when a recovered branch has a support value higher than a given threshold and is present in our reference tree; (2) the false positive rate (or type I error), when a recovered branch has a support value higher than a given threshold and is not present in the reference tree; (3) the true negative rate, when a recovered branch has a support value lower than a given threshold and is not present in the reference tree; and (4) the false negative rate (or type II error), when a recovered branch has a support value lower than a given threshold and is present in the reference tree. To summarize these metrics, we measured the Matthews correlation coefficient (*Matthews, 1975*), also known as the Yule phi ($\varphi$) coefficient (*Yule, 1912*). This metric ranges from a negative one (a total disagreement of the predictions) to a positive one (a perfect agreement of the predictions), measuring the power of the prediction of a binary (true/false) classification estimator.

## Statistical and tree topology tests

We performed a series of statistical analyses to verify the normality (Shapiro–Wilk test; *Shapiro & Wilk, 1965*) and homogeneity of variances of the accuracy and precision (nRF and 1-CFI) (Levene's test; *Levene, 1960*). The difference in performance when comparing different phylogenetic tree-building methods was statistically tested using the four support thresholds (majority-rule consensus = 50%, 65%, 80%, and 95%) by applying general linear model statistical tests and corresponding nonparametric tests for cases lacking normality and/or homogeneity of variances (*Kruskal & Wallis, 1952*; *Nelder & Wedderburn, 1972*).

In addition to testing individual methods, we also checked the performance of groups of methods using an independent-sample Student's t test (*Gosset, 1908*), namely, maximum parsimony methods (EW-MP and IW-MP) *vs* likelihood-based methods (ML and BI). Since all the statistical testing in this study represents a multiple comparison problem, the *Šidák (1967)* correction was applied to control for the familywise error rate (m = 2,

$\alpha' = 0.0253$), *i.e.*, a correction of the *p* value ($\alpha = 0.05$). All the tests described above were performed using Past 4.01 software (*Hammer, Harper & Ryan, 2001*).

We performed a series of classic nonparametric tree topology tests (*i.e.*, paired-site tests *sensu Felsenstein, 2004b*) that can be used in parsimony- and likelihood-based phylogenies (*Felsenstein, 2004b*; *Goldman, Anderson & Rodrigo, 2000*; *Shimodaira, 2002*), following a well-known similar approach used in many studies (*e.g.*, *Buckley, 2002*; *Černý & Simonoff, 2023*; *Ferreira et al., 2023*; *Schneider, 2007*). Our topology tests included both parsimony- and likelihood-based tests to avoid any bias favouring any tested phylogenetic tree-building method.

Two parsimony-based tests were performed: the Templeton test, or Wilcoxon signed-rank test (*Templeton, 1983*), and the Winning-sites test (*Prager & Wilson, 1988*). Three likelihood-based tests were performed: the two-tailed Kishino-Hasegawa test (*Kishino & Hasegawa, 1989*); the one-tailed Shimodaira-Hasegawa test or paired-sites test (*Shimodaira & Hasegawa, 1999*); and Shimodaira's approximately unbiased test (*Shimodaira, 2002*). These likelihood-based tests were performed by applying the bootstrap resampling method with efficient resampling estimated log-likelihood optimization (*Felsenstein, 2004b*; *Kishino, Miyata & Hasegawa, 1990*; *Kishino & Hasegawa, 1989*).

All these tests were applied to all the reconstructed majority-rule consensus trees to avoid inappropriate comparisons of trees with different support thresholds (*Brown et al., 2017*; *Smith, 2019*). Thus, each reconstructed topology was considered a competing phylogenetic hypothesis to be tested. The results of these tests are aligned with those of with tests performed with the same dataset but with optimal point estimate trees. See *Holder, Sukumaran & Lewis (2008)*, *Brown et al. (2017)*, and *O'Reilly & Donoghue (2018)* for more about our preferred use of majority-rule consensus trees. Therefore, we were able to test the congruence of these topological tests with the applied performance indices (accuracy–nRF; and precision, measured *via* resolution–1-CFI) through Pearson (r) and Spearman (ρ) correlations.

With this procedure, we were able to effectively infer whether tree topology tests could be used as indirect measures of phylogenetic performance (*Hillis, 1995*; *Li & Zharkikh, 1995*). This is highly relevant, considering that the application of tree topology tests does not require the knowledge of a known or a well-supported reference tree and therefore can be used in the day-to-day practice of empirical phylogenetic inference to help with decision-making in tree selection. All tree topology tests were performed using PAUP* 4.0a169 (*Swofford & Bell, 2017*). Commands to perform the phylogenetic analyses and topology tests are presented in File S2, and more detailed phylogenetic trees are presented in Files S3–S10.

## RESULTS

### Performance of tree-building methods

In this study, we aim to evaluate the performance of tree-building methods in recovering a reference Hexapoda topology using a morphologic dataset. Among all the tree-building methods and models tested, the trees generated with maximum likelihood (log-likelihood

for Mk = −1,509.9958; log-likelihood for Mk+G = −1,524.3199; Fig. 3) and Bayesian inference (number of trees in the 95% postburn-in credibility interval for Mk = 3,356 and for Mk+G = 3,791; standard deviations of splits <0.006; effective sample size >1,000; potential scale reduction factor ≈1.000) performed better, specifically with higher precision measured *via* resolution (1-CFI) when compared to trees generated *via* maximum parsimony (EW-MP with 252 most parsimonious trees; IW-MP with 24 most parsimonious trees). This was evaluated using a more complex test that properly considers accuracy (nRF) and resolution (1-CFI) together as dependent variables (analysis of covariance F = 7.07, $p$ = 0.00077; Tables 1–4) and when we clustered methods into more inclusive groups, as previously mentioned (analysis of covariance F = 17.61, $p$ = 0.00016; Fig. 4 and Tables 1–4). This difference is also statistically significant when we apply a more straightforward Student's t test (t = 3.8045, $p$ = 0.0005).

Specifically, in the 50% (*i.e.*, majority-rule consensus trees) support threshold, the same topology was recovered with the ML-Mk, BI-Mk, and BI-Mk+G methods and models, and a very similar topology was recovered with ML-Mk+G. (Fig. 3, see Files S3–S10). Among all the phylogenetic tree-building methods tested, maximum parsimony methods exhibited slightly higher accuracy (both RF and nRF) than either the maximum likelihood or Bayesian inference methods, particularly in the 50% support threshold. This difference is *not* statistically significant; therefore, this difference must be interpreted with caution (t = 1.7726, $p$ = 0.0843). Additionally, there was no statistically significant difference in performance (nRF and 1-CFI) between different maximum parsimony methods (EW-MP, IW-MP) (analysis of covariance F = 0.2134, $p$ = 0.6489) or between different likelihood-based methods (ML, BI) (analysis of covariance F = 1.854, $p$ = 0.1965). Finally, there was no statistically significant difference between the Mk and Mk+G models (analysis of covariance F = 1.571, $p$ = 0.2521).

## 50–95% support threshold performance

To evaluate the relationship between support and the known trade-off between phylogenetic accuracy and phylogenetic resolution, we collapsed poorly supported splits of the reconstructed topologies according to four support threshold categories (majority-rule consensus = 50%, 65%, 80%, and 95%). Hence, we compared the support values of accurate ("true") and inaccurate ("false") internal nodes in each threshold category. For all phylogenetic tree-building methods tested, support values for accurately recovered internal nodes (mean = 86.04%, median = 90.11%) were slightly higher (t = 2.0796, $p$ = 0.0376) than those for inaccurate nodes (mean = 82%, median = 86.81%).

Among the accurately recovered internal nodes, the nonparametric bootstrap of maximum parsimony methods had lower (t = 4.9082, $p$ = 0.0001) support values (mean = 83.48%, median = 86.09%) than the maximum likelihood UFBoot or the Bayesian inference PP (mean = 93.33%, median = 98.75%). This pattern repeats itself in all the support thresholds (Fig. 5 and Tables 1–4). Nevertheless, among inaccurately recovered internal nodes, there was no significant difference (t = 0.53957, $p$ = 0.589) between the support values of the nonparametric bootstrap of maximum parsimony methods (mean = 82.43%, median = 91.66%) and those of the maximum likelihood UFBoot and the

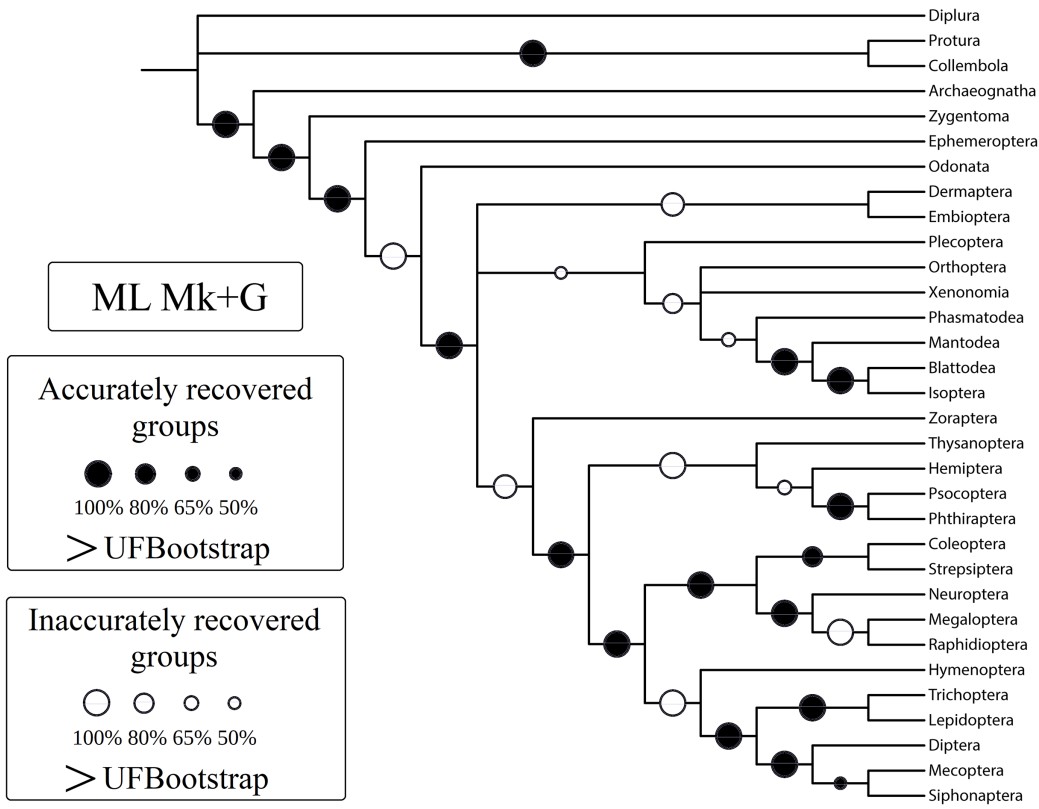

**Figure 3 Cladogram (reanalysis of the Beutel & Gorb).** Cladogram resulting from the reanalysis of the *Beutel & Gorb (2001)* matrix performing maximum likelihood (ML) with the Mk+G model. Groups with support ("ultrafast" bootstrap, UFBoot) values less than 50% were collapsed. A very similar topology was recovered with ML-Mk, BI-Mk, and BI-Mk+G (see Files S3–S10). All four cladograms (ML-Mk, ML-Mk +G, BI-Mk, and BI-Mk+G) presented an optimal trade-off between accuracy and precision (measured *via* resolution) (nRF = 0.392; 1-CFI = 0.068). Nodes with black circles represent accurately recovered groups, and nodes with white circles represent inaccurately recovered groups (when compared to the well-- supported reference tree, *Misof et al. (2014)*).

Bayesian inference PP (mean = 84.53%, median = 91.60%). This pattern is also seen in all the support thresholds (Fig. 5 and Tables 1–4).

The true positive rate (power), the false positive rate (type I error), the true negative rate, the false negative rate (type II error), and the Matthews correlation coefficient did not present statistically significant differences among the methods (all analysis of variance F < 2.366, all $p$ > 0.09398; all Kruskal–Wallis Hc < 5.814, all $p$ > 0.121). When the methods were clustered, the true positive rate of UFBoot for maximum likelihood and posterior probability for Bayesian inference presented values slightly higher ($p$ < 0.05) than those of the nonparametric bootstrap for maximum parsimony methods.

Although this difference was significant in a Student's t test (t = 2.0842, $p$ = 0.0439), this result must be interpreted with caution since this value is close to α = 0.05 and above the value of the *Šidák (1967)* correction (α′ = 0.0253). Finally, all values of the Matthews correlation coefficient were positive (mean = 0.3897, median = 0.4329), and all tree-building methods, among all support thresholds applied, presented a comparable general performance (t = 1.7297, $p$ = 0.0947) concerning the power of the prediction of

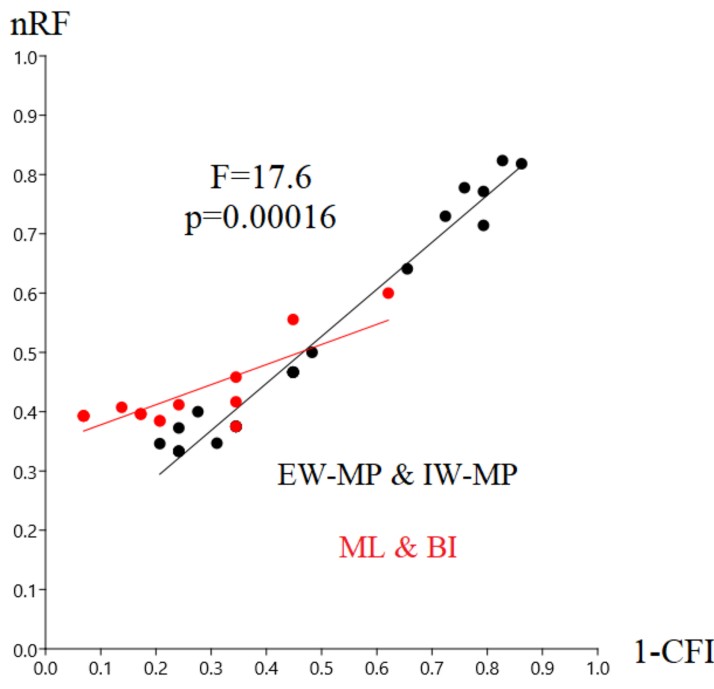

**Figure 4 Differences in nRF and precision.** Differences in the normalized Robinson-Foulds metric (nRF) and precision, measured *via* resolution (one minus Colless' consensus fork index, 1-CFI) among the performed phylogenetic tree-building methods. All four different values for collapsing groups (support thresholds) were considered (MRC – 50%, 65%, 80%, and 95%). Reanalyses were performed using the *Beutel & Gorb (2001)* matrix. Phylogenetic tree-building methods considered: equal-weights maximum parsimony–EW-MP; implied-weights maximum parsimony–IW-MP; maximum likelihood–ML; and Bayesian inference–BI.

**Table 1 Absolute number of recovered splits and accurately recovered splits (50%).** Absolute number of recovered splits and accurately recovered splits (50%). Mean (%) of the support values (non-parametric bootstrap; "ultrafast" bootstrap, UFBoot; and posterior probability, PP) of the recovered splits, accurately recovered splits, and inaccurately recovered splits. Accuracy, measured *via* the normalized Robinson-Foulds metric (nRF), and precision, measured *via* resolution (one minus Colless' consensus fork index, 1-CFI). Groups were collapsed using support thresholds of 50%.

|  | Support mean (accurate splits) | Support mean (inaccurate splits) | nRF | 1-CFI |
|---|---|---|---|---|
| EW-MP | 78.76 | 78.20 | 0.3725 | 0.2414 |
| IW-MP K = 2 | 85.55 | 85.62 | 0.3333 | 0.2414 |
| IW-MP K = 3 | 84.64 | 85.45 | 0.3333 | 0.2414 |
| IW-MP K = 5 | 84.21 | 80.34 | 0.3462 | 0.2069 |
| IW-MP K = 10 | 82.69 | 86.47 | 0.3333 | 0.2414 |
| IW-MP K = 20 | 84.93 | 80.21 | 0.4000 | 0.2759 |
| ML Mk | 90.18 | 79.50 | 0.3929 | 0.0690 |
| ML Mk+G | 96.06 | 84.20 | 0.3929 | 0.0690 |
| BI Mk | 93.37 | 88.78 | 0.3929 | 0.0690 |
| BI Mk+G | 93.69 | 86.24 | 0.3929 | 0.0690 |

**Table 2 Absolute number of recovered splits and accurately recovered splits (65%).** Absolute number of recovered splits and accurately recovered splits (65%). Mean (%) of the support values (non-parametric bootstrap; "ultrafast" bootstrap, UFBoot; and posterior probability, PP) of the recovered splits, accurately recovered splits, and inaccurately recovered splits. Accuracy, measured *via* the normalized Robinson-Foulds metric (nRF), and precision, measured *via* resolution (one minus Colless' consensus fork index, 1-CFI). Groups were collapsed using support thresholds of 65%.

| | Support mean (accurate splits) | Support mean (inaccurate splits) | nRF | 1-CFI |
|---|---|---|---|---|
| EW-MP | 84.81 | 86.83 | 0.4667 | 0.4483 |
| IW-MP K = 2 | 87.39 | 92.64 | 0.3469 | 0.3103 |
| IW-MP K = 3 | 88.08 | 92.40 | 0.3750 | 0.3448 |
| IW-MP K = 5 | 87.81 | 92.21 | 0.3750 | 0.3448 |
| IW-MP K = 10 | 86.83 | 92.62 | 0.3750 | 0.3448 |
| IW-MP K = 20 | 84.93 | 91.01 | 0.3750 | 0.3448 |
| ML Mk | 92.13 | 86.00 | 0.3962 | 0.1724 |
| ML Mk+G | 98.63 | 94.71 | 0.3846 | 0.2069 |
| BI Mk | 95.91 | 93.00 | 0.4074 | 0.1379 |
| BI Mk+G | 96.13 | 93.84 | 0.3962 | 0.1724 |

**Table 3 Absolute number of recovered splits and accurately recovered splits (80%).** Mean (%) of the support values (non-parametric bootstrap; "ultrafast" bootstrap, UFBoot; and posterior probability, PP) of accurately recovered splits and inaccurately recovered splits. Accuracy, measured *via* the normalized Robinson-Foulds metric (nRF), and precision, measured *via* resolution (one minus Colless' consensus fork index, 1-CFI). Groups were collapsed using the support threshold of 80%.

| | Support mean (accurate splits) | Support mean (inaccurate splits) | nRF | 1-CFI |
|---|---|---|---|---|
| EW-MP | 94.03 | 91.32 | 0.6410 | 0.6552 |
| IW-MP K = 2 | 92.52 | 92.64 | 0.4667 | 0.4483 |
| IW-MP K = 3 | 92.02 | 92.40 | 0.4667 | 0.4483 |
| IW-MP K = 5 | 91.73 | 92.21 | 0.4667 | 0.4483 |
| IW-MP K = 10 | 90.75 | 92.62 | 0.4667 | 0.4483 |
| IW-MP K = 20 | 89.53 | 91.01 | 0.5000 | 0.4828 |
| ML Mk | 94.21 | 94.00 | 0.4167 | 0.3448 |
| ML Mk+G | 98.53 | 94.71 | 0.3846 | 0.2069 |
| BI Mk | 95.91 | 95.54 | 0.3962 | 0.1724 |
| BI Mk+G | 97.65 | 97.58 | 0.4118 | 0.2414 |

phylogenetic groups. The receiver operating characteristic plots of the true positive rate (power) and the false positive rate (type I error) can be seen in Fig. 6.

## Tree topology tests

To estimate the performance of tree topology tests, the test probability value for a given reconstructed tree was compared to the accuracy and precision of the tree, such that more appropriate tests were those that yield a higher probability value for a reconstructed tree

**Table 4 Absolute number of recovered splits and accurately recovered splits (95%).** Mean (%) of the support values (non-parametric bootstrap; "ultrafast" bootstrap, UFBoot; and posterior probability, PP) of accurately recovered splits and inaccurately recovered splits. Accuracy, measured *via* the normalized Robinson-Foulds metric (nRF), and precision, measured *via* resolution (one minus Colless' consensus fork index, 1-CFI). Groups were collapsed using the support threshold of 95%.

| | Support mean (accurate splits) | Support mean (inaccurate splits) | nRF | 1-CFI |
|---|---|---|---|---|
| EW-MP | 99.67 | 95.50 | 0.8182 | 0.8621 |
| IW-MP K = 2 | 98.55 | 96.03 | 0.7297 | 0.7241 |
| IW-MP K = 3 | 98.20 | 97.13 | 0.7143 | 0.7931 |
| IW-MP K = 5 | 98.86 | 95.72 | 0.7714 | 0.7931 |
| IW-MP K = 10 | 98.97 | 95.40 | 0.7778 | 0.7586 |
| IW-MP K = 20 | 99.77 | 95.75 | 0.8235 | 0.8276 |
| ML Mk | 98.25 | 97.67 | 0.6000 | 0.6207 |
| ML Mk+G | 99.87 | 100.00 | 0.3750 | 0.3448 |
| BI Mk | 99.29 | 98.96 | 0.5556 | 0.4483 |
| BI Mk+G | 98.95 | 98.86 | 0.4583 | 0.3448 |

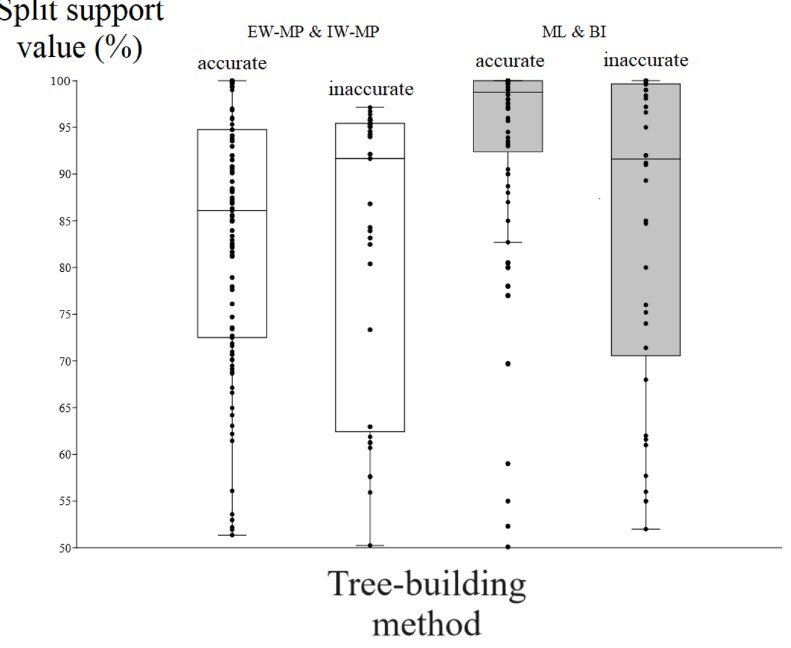

**Figure 5 Differences in split support values.** Differences (median with confidence interval) in split support values (non-parametric bootstrap; "ultrafast" bootstrap, UFBoot; and posterior probability, PP) among accurately recovered splits and inaccurately recovered splits. Groups were collapsed using support thresholds of 50%. Phylogenetic tree-building methods considered: equal-weights maximum parsimony–EW-MP; implied-weights maximum parsimony–IW-MP; maximum likelihood–ML; and Bayesian inference–BI.

more similar to our reference tree. As likelihood-based topologies were more similar to our reference tree, that is ML and BI methods performed best, we evaluated tests on their ability to yield higher probabilities for the ML and BI (reconstructed) topologies than for

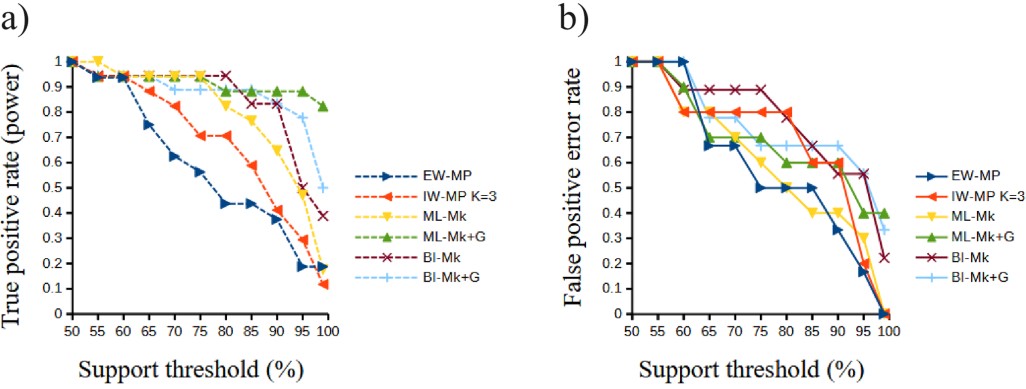

**Figure 6 Receiver operating characteristic plots.** Receiver operating characteristic plots. (A) True positive rate (power) related to the support thresholds applied; (B) false positive rate (type I error) related to the support thresholds. Phylogenetic tree-building methods considered: equal-weights maximum parsimony–EW-MP; implied-weights maximum parsimony–IW-MP K = 3; maximum likelihood, k-state Markov model without a discrete gamma distribution–ML-Mk; maximum likelihood, k-state Markov model with a discrete gamma distribution–ML-Mk+G; Bayesian inference, k-state Markov model without a discrete gamma distribution–BI-Mk; and Bayesian inference, k-state Markov model with a discrete gamma distribution–BI-Mk+G.               

those reconstructed using maximum parsimony methods. All tests yielded higher probabilities for the reconstructed trees using ML and BI (Templeton's ML and BI $p > 0.7389$; Winning-sites's ML and BI $p = 1.00$; Kishino-Hasegawa ML and BI $p > 0.7219$; Shimodaira-Hasegawa ML and BI $p > 0.6589$; Shimodaira's approximately unbiased $p > 0.3988$) than values for the MP topologies (Templeton's MP $p < 0.023$; Winning-sites's MP $p < 0.031$; Kishino-Hasegawa MP $p < 0.052$; Shimodaira-Hasegawa MP $p < 0.036$; Shimodaira's approximately unbiased $p < 0.0063$) (see Table 5 for more details).

Our results suggest that these topology tests are reliable estimators of phylogenetic performance. More specifically, our recommendation is supported by the fact that the precision, measured *via* resolution (1-CFI), was highly correlated to all the applied tree topology tests: Templeton (Pearson $R^2 = 0.940$, Spearman $R^2 = 0.924$, linear regression $p = 0.0001$); Winning sites (Pearson $R^2 = 0.972$, Spearman $R^2 = 0.960$, linear regression $p = 0.0004$); Kishino and Hasegawa (Pearson $R^2 = 0.948$, Spearman $R^2 = 0.924$, linear regression $p = 0.0001$); Shimodaira and Hasegawa (Pearson $R^2 = 0.973$, Spearman $R^2 = 0.924$, linear regression $p = 0.0001$); and Shimodaira's approximately unbiased (Pearson $R^2 = 0.946$, Spearman $R^2 = 0.891$, linear regression $p = 0.0004$) (see File S12).

## DISCUSSION

The most important finding of our study is that, based on an empirical discrete morphological dataset of hexapods, likelihood-based methods can build trees with better performance (specifically, better precision, measured *via* resolution) than maximum parsimony methods. In the dataset used (*Beutel & Gorb, 2001*), both maximum likelihood and Bayesian inference methods were equally effective. Additional studies with other datasets are needed to further explore the application of the pattern that was found in the present study to other lineages. Assuming a well-supported reference tree

**Table 5  Topological tests.**

|  | ln(L) | Templeton | Winning-sites | KH | SH | AU |
|---|---|---|---|---|---|---|
| EW-MP | −1,147.60 | 0.0062 | 0.0039 | 0.0135 | 0.0152 | 0.0000 |
| IW-MP K = 2, 3, 10 | −1,161.77 | 0.0042 | 0.0020 | 0.0069 | 0.0072 | 0.0000 |
| IW-MP K = 5 | −1,143.78 | 0.0231 | 0.0313 | 0.0521 | 0.0359 | 0.0063 |
| IW-MP K = 20 | −1,152.22 | 0.0024 | 0.0010 | 0.0058 | 0.0070 | 0.0000 |
| ML Mk | −1,121.02 | 0.7389 | 1.0000 | 1.0000 | 0.7472 | 0.5376 |
| ML Mk+G | −1,123.58 | 1.0000 | 1.0000 | 0.7219 | 0.6940 | 0.3988 |
| BI Mk | −1,121.02 | 0.7389 | 1.0000 | 1.0000 | 0.7472 | 0.5393 |
| BI Mk+G | −1,121.02 | 0.7389 | 1.0000 | 1.0000 | 0.7472 | 0.5393 |

**Note:**
Performed topology tests: Templeton test; Winning-sites test; Kishino-Hasegawa (KH) test; weighted Kishino-Hasegawa (WKH) test; Shimodaira-Hasegawa (SH) test; weighted Shimodaira-Hasegawa (WSH) test; and Shimodaira's approximately unbiased (AU) test. The presented numbers are the p values of each corresponding statistic. All tests were applied to all the reconstructed majority-rule consensus trees (MRC); thus, each topology was considered a competing phylogenetic hypothesis. The results of these tests agree with tests performed with the same dataset but with optimal point estimate trees (see *Brown et al. (2017)* for more on the subject of our preferred use of majority-rule consensus trees). Phylogenetic tree-building methods considered: equal-weights maximum parsimony–EW-MP; implied-weights maximum parsimony–IW-MP; maximum likelihood–ML; Bayesian inference–BI. ln(L)–log-likelihood of reconstructed majority-rule consensus trees (MRC).

(*Misof et al., 2014*), these k-state Markov (Mk) analyses resulted in unexpectedly high precision measured *via* resolution (1-CFI = 0.069).

The results of the present study are aligned with the findings presented in recent works focusing primarily on simulations of discrete morphological data (*Brown et al., 2017*; *O'Reilly et al., 2016*, *2017*, *2018*; *Puttick et al., 2017a*, *2017b*, *2018*; *Wright & Hillis, 2014*) and are in agreement with a previous study based on experimental data that likewise indicates the superior performance of maximum likelihood and Bayesian inference methods (*Randall et al., 2016*). Some researchers (*Goloboff & Arias, 2019*; *Goloboff, Torres & Arias, 2017*; *Goloboff et al., 2019*) have questioned the reported (*O'Reilly et al., 2016*, *2017*, *2018*; *Puttick et al., 2017a*, *2017b*, *2018*) lower performance (accuracy, precision, or both) of maximum parsimony (EW-MP and IW-MP) when compared to likelihood-based methods. They argue that differences in branch lengths, especially in the deep nodes of asymmetric trees, would artificially generate a bias in favour of likelihood-based methods. Branch lengths and other parameters can be controlled in simulation studies but not in empirical studies such as ours. In any case, this would not be a problem in our study, as maximum parsimony and likelihood-based methods recovered the deep nodes similarly; see, for instance, the early Hexapoda splits, which were unresolved, and the accurate paraphyletic position of the Apterygota orders (Figs. 1 and 3).

Comparatively, equal-weights maximum parsimony can be interpreted as a parameter-rich evolutionary model (*Goldman, 1990*; *Penny et al., 1994*; *Tuffley & Steel, 1997*). In addition to the problems associated with overparameterization (see *Holder, Lewis & Swofford, 2010*; *Huelsenbeck et al., 2008*; *Huelsenbeck, Alfaro & Suchard, 2011*), this class of methods (in this statistical interpretation or not) has been historically criticized as sensitive to long-branch attraction artefacts (*e.g.*, *Allard & Miyamoto, 1992*; *Carmean & Crespi, 1995*; *Felsenstein, 1978*; *Huelsenbeck, 1995*), in which parsimony can be biased in certain specific combinations of branch lengths. This situation was assessed not only in

molecular but also in morphological datasets (*Lee & Worthy, 2012*; *Lockhart & Cameron, 2001*; *Wiens & Hollingsworth, 2000*). This finding is in apparent agreement with our result, but it would be best to further explore the relationship between long-branch attraction artefacts and the phylogenetic performance measures tested in this study.

Likelihood-based methods have also been shown to be biased when applied to other branch-length combinations (*Kück et al., 2012*; *Susko, 2012*), particularly in the presence of heterotachy (*Zhou et al., 2007*), *i.e.*, when the evolutionary rate of a given site/character varies across time/phylogenetic history (*Philippe et al., 2005*; *Kolaczkowski & Thornton, 2004*). Furthermore, under certain conditions, long-branch attraction artefacts were also effectively demonstrated in Bayesian inference (*Kolaczkowski & Thornton, 2009*; *Susko, 2012*). Since most of these studies focus on simulations, attempts to assess method performance in empirical data, such as ours, are important for a better understanding of these long-branch artefacts.

Additionally, it has been reported that models incorporating heterotachy in the evolutionary process of the dataset performed significantly better than traditional evolutionary models (*Kolaczkowski & Thornton, 2008*). Unfortunately, such complex models have not yet been efficiently implemented for morphological data. Future studies should focus on the impact of using alternative and more complex models on morphological data and how to incorporate the complexity of the morphological change in evolutionary models (see also *Keating et al., 2020*).

As we have shown, likelihood-based methods present better precision than maximum parsimony methods, or they at least show comparable performance. In this sense, a preference for likelihood-based methods has been suggested, as they incorporate branch length information and maximum parsimony does not (*Felsenstein, 1973*, *1978*, *1981*). Thus, likelihood-based methods are, by definition, more informative methods of phylogenetic reconstruction. It is also worth mentioning that the tested phylogenetic tree-building methods are not theoretically limited to one type of data (*Edwards, 1996*, *2009*; *Felsenstein, 2001*, *2004c*; *Sober, 2004*). This interpretation agrees with the findings of our study since we have shown that maximum likelihood and Bayesian inference methods can outperform maximum parsimony methods for morphological-based phylogenies, at least for those among hexapods.

As the interpretation of the accuracy and resolution of a phylogenetic tree depends on the support value, we also detailed our results using "extreme" support thresholds to evaluate reliable splits, namely, those with 80% and 95% threshold values. In most cases, similar performance values (nRF and 1-CFI) were found in likelihood-based methods (Tables 3 and 4). Considering the threshold of 95%, for example, for the maximum likelihood trees with the Mk+G model, reconstruction performed significantly better (nRF = 0.375 and 1-CFI = 0.344) than that with the Mk model (nRF = 0.600 and 1-CFI = 0.620). The same pattern was observed for BI with a support threshold of 95%.

The results reported here indicate that among the accurately recovered internal nodes, nonparametric bootstrapping of maximum parsimony methods had lower support values than the maximum likelihood UFBoot or PP of Bayesian inference. This pattern strongly agrees with the interpretation that resampling methods (nonparametric bootstrap and

jackknife) applied to maximum parsimony are more conservative and tend to underestimate support if compared to Bayesian PP, a more liberal measure that often overestimates support. This interpretation, involving statistical support measures, was demonstrated in simulated and empirical datasets, and it has been well known for quite some time and has been explored in the phylogenetic literature (*e.g.*, *Anisimova & Gascuel, 2006*; *Anisimova et al., 2011*; *Hillis & Bull, 1993*; *Wilcox et al., 2002*). Additionally, this pattern indicates a high statistical power (*Anisimova et al., 2011*) among the support thresholds of likelihood-based methods applied in the present study (UFBoot and PP) when compared to the nonparametric bootstrap of maximum parsimony.

Previous studies have explored the possibility of using the Templeton, Kishino and Hasegawa, Shimodaira and Hasegawa, and Swofford–Olsen–Waddell–Hillis (SOWH) tests (the latter is a complex test applied *via* parametric bootstrapping; see *Goldman, Anderson & Rodrigo, 2000*; *Hillis, Mable & Moritz, 1996*; *Swofford et al., 1996*) as indirect estimators of several performance measures for selecting competing alternative topologies (*i.e.*, specific phylogenetic hypotheses) and genetic markers (*e.g.*, *Hipp, Hall & Sytsma, 2004*; *Miya & Nishida, 2000*; *Rokas et al., 2002*; *Zardoya & Meyer, 1996*). In particular, the Shimodaira's approximately unbiased test has been previously recommended by others as the least biased among tree topology tests (*Shimodaira, 2002*; *Swofford & Bell, 2017*).

Our results indicate that these tree topology tests are reliable estimators of phylogenetic performance to be used when selecting between alternative trees. More specifically, our recommendation is strongly supported by the fact that the precision, measured *via* resolution (1-CFI), highly correlates with all the tree topology tests applied: Templeton, Winning-sites, Kishino and Hasegawa, Shimodaira and Hasegawa, and Shimodaira's approximately unbiased (see *Ferreira et al., 2023*; *Goldman, Anderson & Rodrigo, 2000*; *Schneider, 2007* for guidance).

## CONCLUDING REMARKS

Using a morphological dataset, for the first time, our study suggests that likelihood-based methods build more precise phylogenies than maximum parsimony methods, at least when applied to our hexapod data. Despite many claims that molecular sequence data have replaced morphological datasets in phylogenies, our study highlights the existence of a fair phylogenetic signal in a morphological dataset when recovering a phylogenomic tree of hexapods. This an important result, considering that over 99% of the biodiversity in our planet is fossil and only the morphology is accessible to phylogenies using those taxa. This finding reinforces the view that classic morphological phylogenetic analyses and other tests of morphological based methods are still much necessary.

## ACKNOWLEDGEMENTS

We are very grateful to Rolf G. Beutel (Friedrich-Schiller-Universität Jena) for kindly making their data matrix available and for having provided important articles and suggested additional references. We are also grateful to André Silva Roza, Beatriz Mello Carvalho, Carlos Eduardo Guerra Schrago, Elliot Santovich Scaramal (Universidade Federal do Rio de Janeiro, UFRJ), Les R. Foulds (Universidade Federal de Goiás, UFG),

Mario César Cardoso de Pinna (Museu de Zoologia da Universidade de São Paulo, MZUSP), and Vinicius de Souza Ferreira (Natural History Museum of Denmark, Zoological Museum, University of Copenhagen) for revising a preliminary version of the text.

### Funding

This work was supported by the Coordenação de Aperfeiçoamento de Pessoal de Nível Superior–Education Ministry of Brazil (CAPES)–Finance Code 001. The National Research and Technology Council (CNPq) process 312786/2022-0 provided support to Jose Ricardo Miras Mermudes and 310567/2018-1 to Claudia AM Russo. The Rio de Janeiro State Research Funding Agency (FAPERJ) processes E-26/010.001887/2019, SEI-260003/001170/2020, SEI-260003/012995/2021 provided support to Claudia AM Russo. The funders had no role in study design, data collection and analysis, decision to publish, or preparation of the manuscript.

### Grant Disclosures

The following grant information was disclosed by the authors:
Coordenação de Aperfeiçoamento de Pessoal de Nível Superior–Education Ministry of Brazil (CAPES): 001.
The National Research and Technology Council (CNPq): 312786/2022-0 & 310567/2018-1.
The Rio de Janeiro State Research Funding Agency (FAPERJ): E-26/010.001887/2019, SEI-260003/001170/2020 & SEI-260003/012995/2021.

### Competing Interests

The authors declare that they have no competing interests.

### Author Contributions

- Felipe Francisco Barbosa conceived and designed the experiments, performed the experiments, analyzed the data, prepared figures and/or tables, authored or reviewed drafts of the article, and approved the final draft.
- José Ricardo M. Mermudes conceived and designed the experiments, analyzed the data, authored or reviewed drafts of the article, and approved the final draft.
- Claudia A. M. Russo conceived and designed the experiments, analyzed the data, authored or reviewed drafts of the article, and approved the final draft.

### Data Availability

The modifications to the morphological matrix, commands, reference tree and results in detail are available in the Supplemental Files.

## Supplemental Information

Supplemental information for this article can be found online at http://dx.doi.org/10.7717/peerj.16706#supplemental-information.

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
