# Peer review of "Performance of tree-building methods using a morphological dataset and a well-supported Hexapoda phylogeny"

_PeerJ, doi:10.7717/peerj.16706_

## Round 0.1 · original submission · Major Revisions

I think this manuscript represents an interesting approach to evaluating the efficacy and analytical approaches to morphological characters in hexapods - likely extensible to other organismal groups (applicable to these types of analyses in general). However, the reviewers identify a number of concerns, particularly focusing on clarity of experimental design (choice of data sets etc.). Please pay particular attention to the views expressed by Reviewer 3. I may likely send the manuscript out for review again after it has been revised.

·

Basic reporting

In general, the manuscript is well written and clear; however, some aspects in the introduction need to be supplemented
a) For example, after line 98, Although it is well justified the Performance of phylogenetic methods using discrete morphological data, the phylogenetic inconsistencies presented in hexapods are not sufficiently justified, depending on the phylogenetic method and/or the empirical data.
b) Which are these?
c) what impact would the type of data set have on the phylogenetic reconstruction of hexapoda?
d) in this context, in line 110, it was not clear how the phylogeny of Figure 1 justifies new phylogenetic analyses in hexapoda. Also, is this phylogeny only from Misof et al., 2014 or is it a consensus of Misof et al., 2014, Beutel et al., 2017 and Kjer et al., 2016?

Experimental design

Despite presenting an original and modern methodology, relevant methodological aspects should be improved in the Materials & Methods section
A) It is not clear why the authors make a discussion of the inconsistencies between the phylogenies of various authors with the one used as a reference (lines 134 to 153). 1) this should be in the discussion of results and, 2) it generates the doubt that the phylogeny selected as reference is not robust enough as "reference" for all hexapoda, generating from the beginning a possible bias in the analyses. this main point should be sufficiently clarified, in order to support the results and conclusions of the work.
B) It is not clear why the authors selected the Beutel & Gorb 2001 dataset. Please explain why you used this only one data set. There are now more organisms sampled, classified and catalogued. therefore, these could be biased in several taxonomic groups of little medical or economic importance. how do the authors support this? for example, if another, more current dataset was used, could the same phylogenetic relationships be generated?
C) the other methodologies applied by the authors are adequate and well described.

Validity of the findings

The methodological aspects must be clarified (see previous comments) for the results to be validated.

Additional comments

In general, the manuscript is well written and clear; however, key aspects in the Materials & Methods section need to be supplemented

·

Basic reporting

The English is good and the background very solid, including the very good coverage of relevant literature.
The study is clearly structured and well-illustrated.
The presented data are relevant for the discussed hypotheses.

Experimental design

the analytical methods applied are impeccable. This part of the study is very convincing.

Validity of the findings

It is convincingly shown that Ml and BI outperform MP and that these methods should be preferred in analyses of morphological characters. I fully agree with the argumentation and conclusions.

Additional comments

This study addresses a very important topic in phylogenetics, the performance of different analytical methods. The applied data and analytical methods are sound. The authors are obviously very familiar with current analytical approaches.
l. 91. Hennig’s approach was “Phylogenetische Systematic” (= phylogenetic systematics), a non-numerical evaluation of morphological characters, based on the concept of synapomorphy. This implied stepwise tree building without a formal analyses and without explicit outgroup comparison for assessment of character state polarity. Hennig did not explicitly use the concept of parsimony. This “evolved” and became known as cladistics, with numerical analyses of character state matrices and outgroups used for rooting trees.
The authors used the morphological matrix from Beutel & Gorb (2001). Arguably the updated version from Beutel & Gorb (2006) would have been better, with Mantophasmatodea included. However, this is not crucial for the purpose of the study.
Rolf Beutel

Reviewer 3 ·

Basic reporting

I found the manuscript "Performance of phylogenetic methods using discrete morphological data and a well-supported Hexapoda phylogeny" to be an interesting and original attempt at addressing a hard issue in phylogenetic analysis. This paper is self contained, mostly well written in unambiguous technical English, although some parts of the methods section are hard to follow. Overall, I think the bibliographic review was adequately used to provide sufficient background, but some references (e.g. Rosa et al., 2019) were miscited. The paper is also well structured with sufficient figures and tables, but I could not find the morphological matrix used in the analyses, which I expected to be shared in a suitable format (e.g. NEXUS) as SM. In spite of its qualities, I cannot recommend this manuscript for publication because I believe it fails to address its central point. Although it is certainly hard to simulate data that would adequately emulate morphological evolution, the main point of a simulation is that the "true" phylogeny is known. By employing the 50% majority rule consensus tree in Misof et. al. (2014) as the "true" phylogeny, I believe that the authors have fallen short of meeting  adequate standards that justify their bold claims throughout the manuscript. At most, they have demonstrated that the application of a limited number of likelihood-based methods to a single morphological matrix outperforms a limited number of analyses employing parsimony when it comes to recovering some of the clades with posterior probability equal or higher than 50% reported in Misof et al. (2014). In my view, their design does not allow for much extrapolation beyond this limited statement.

Experimental design

The manuscript addresses a valid point, namely, that simulated data may not be the best way to evaluate the relative performance of alternative methods for phylogenetic reference when it comes to morphological data. This limitation is hard to address given our current knowledge of how those characters evolve and it is an investigation topic well suited to PeerJ. The methods were sufficiently described and results would be replicable if the original matrix was provided. As stated in the previous section, there are fundamental flaws in their reasoning. Every phylogeny is a hypothesis about the evolutionary history of any given group and must be treated as such. While there is no way of knowing the true phylogeny of a clade with relatively low rates of evolution (such as the Hexapoda), some phylogenetic relationships are recurrently recovered in myriads of analyses based on datasets from different sources, even when analyzed under different criteria. In time, these clades become "canonical", in the sense that the rejection of one of them by any given analysis raises suspicions about the methodological integrity of the very analysis, and not about the existence of that clade itself. Hence, I believe that a valid test using "real" data would require several morphological datasets, collected for different groups of organisms, and the statistical demonstration that likelihood-based methods consistently outperform parsimony across those datasets. The authors could claim that the well-supported clade in Misof et al. (2014) have been recurrently recovered in subsequent studies. While this is certainly true, the authors must also keep in mind the limitations of their sampling design when building their conclusions: out of the inumerable, well established relationships in the whole Tree of Life, they have selected a few i.e, those within Hexapoda; out of the several different ways of analyzing the data and the several ways of parameterizing those analyses (convexity values, prior choices, partitioning schemes, rate variation modelling, searching algorithms, etc.) they have arbitrarily chosen a subset and, out of the large universe of potential morphological matrices, they have chosen only one. While an exhaustive survey of this multivariate space is certainly unattainable, greater sampling effort is certainly possible. I refer the authors to April Wright’s PhD dissertation for an example of such effort (https://repositories.lib.utexas.edu/handle/2152/30934).

Validity of the findings

As stated in the previous sections, the manuscript experimental design does not warrant the bold claims in the manuscript’s conclusions and narrowing the scope of these claims would render the paper unsuitable for publication. I also think that there is an excessive number of statistical tests that aggregate little to the paper and make the methods section excessively long and hard to follow. Given the rationale laid out by the authors, I would favor an approach in which different analytical criteria were treated as classifiers, able to identify the well-supported nodes in the reference (“true” or “cannonical”) phylogenies. Using this logic, true positives are the nodes in the reference trees that are also present in the inferred trees; false positives would be the nodes recovered in inferred trees that are absent from the reference phylogenies and false negatives would be the reciprocal cases (i.e. absent from inferred, but present in reference trees). Finally, true negatives would be those nodes absent from both the inferred and reference trees. This latter category may at first sound nonsensical if one is thinking about a single inferred topology. However, the authors did take in account support values in their approach, which leads us to the possibility of constructing Receiver-Operating Characteristic (ROC) curves by varying support threshold (for either PP or bootstrap values). As the threshold increases, nodes whose support fell below the cutoff value and that are not found in the reference tree, would be scored as true negatives. This approach allows for the calculations of ROCs’ Areas Under the Curve (AUCs) or alternative, less biased performance metrics, such as True Skill Statistic (TSS). Additionally, it yields more standard computation of statistics such as accuracy and precision. I refer the authors to Anisimova et al. (2011) for an example of this approach applied to evaluation of support methods in phylogenetic inference (https://academic.oup.com/sysbio/article/60/5/685/1644562).

Reviewer 4 ·

Basic reporting

Overall, the manuscript would benefit from substantial revisions to meet publication standards. Its current writing style is convoluted and tends to be verbose, hindering clarity and objectivity. Certain passages resemble textbook explanations and feel out of place in a research paper.
- The introduction needs better structuring. As it stands, it seems to be a blend of broad statements combined with a narrow case study, punctuated with unrelated historical details. A substantial restructure is necessary for a more objective presentation. There is excessive referencing to Hillis (1995) and Miyamoto and Fitch (1995). The first two paragraphs come across as verbose and unclear. English editing is essential. For instance:
1) Some statements are formulated awkwardly, such as "few researchers seem eager to discuss the performance of current phylogenetic methods that use few genetic markers or phylogenomics." Are you suggesting that methodological performance is not assessed on multi-loci datasets? This is certainly not true.
2) The writing style includes unnecessary details ("see Materials and Methods for more information about the reference tree and certain disagreements").
- The methods section is too wordy and could be cut by over 50%. It's not clear or straightforward, and it reads like a mix of a textbook, review, and research paper. This affects the overall readability of the manuscript. Please revise this section for conciseness and clarity. For example, there's no need for five paragraphs to mention the use of the normalized RF distance. Just state it and offer a brief explanation if you feel it's essential. Refrain from including textbook-like descriptions in a research paper.
- The authors postulate that the Mk model is, in essence, an extrapolation of the Jukes-Cantor paradigm. This assertion holds truth only to a certain degree. Within the JC model, the matrix dimensionality remains at 4x4 across nucleotide sequence alignment sites. Contrarily, when ascertaining the likelihood within morphological matrices, this dimensionality exhibits variability—transitioning from 2x2 for binary characters to higher dimensionalities. This is not exactly JC69.
- Please re-write your results section avoiding citations. The results should focus on communicating your findings and nothing else. The first subsection of Results (Hexapoda phylogeny…) could be excluded.
- In the Results section, the authors delve into commentary, elongating the text unnecessarily. For the sake of objectivity, please strictly present the outcomes, reserving any interpretative comments for the discussion section. For instance, affirming that “There are several publications that can provide some useful guidance on the applications of topology tests” is simply misplaced.

Experimental design

- The primary aim of the authors remains ambiguous: Is it a broad evaluation of tree building methods or a focused study on hexapod phylogeny? Pursuing both goals might not be feasible. I concur that at present, fossil data might be the primary reason for using morphological characters in phylogenetic inference. Given that current insect diversity is just a fraction of what's documented in fossils, I recommend the authors frame their study in this context.
- I suggest excluding the analysis of partitions. The total number of characters (115) is too restrictive to be further subdivided into small partitions which will increase the sampling error. What is the number of characters per partition? It does not bring any relevant information here.

Validity of the findings

- It seems that the major findings are valid, but the manuscript is poorly structured and cannot be published without substantial editing.

---

## Round 0.2 · accepted · Accept

I sincerely appreciate the careful attention to the reviewers' comments. I believe the authors have thoroughly addressed all of the criticisms in the previous review and the manuscript is now suitable for publication. I very much look forward to seeing the paper published - it's a very nice contribution that explores the utility of morphological data (more often than not, now overlooked).